# Development of Universal Mould Geometry for the Teeming of Cylindrical Iron-Base Alloy Ingots

Josef Odehnal [1], Pavel Ludvík [2], Tomáš Studecký [2,*] and Pavel Michálek [2]

1   Department of Material Science and Technology, Faculty of Mechanical Engineering,
    University of West Bohemia, UWB Plzeň, Univerzitní 22, 301 00 Pilsen, Czech Republic;
    odehnal@kmm.zcu.cz
2   COMTES FHT Inc., Průmyslová 995, 334 41 Dobřany, Czech Republic; pavel.ludvik@comtesfht.cz (P.L.);
    pavel.michalek@comtesfht.cz (P.M.)
*   Correspondence: tomas.studecky@comtesfht.cz; Tel.: +420-377-197-333

**Abstract:** The presented work is aimed at developing a mould geometry suitable for casting both low- and high-alloy steel grades into 500 kg experimental ingots. The high Height-to-Diameter (H/D)-ratio mould currently used in COMTES FHT Inc. served as a reference and for finite element method simulations (FEM) of the filling and solidification process. The optimized mould geometry, balancing the porosity and segregations, was determined using MAGMA software. Four different steel grades were defined for the simulation. Case studies were carried out for 34CrNiMo6 (W.Nr. 1.6582), DHQ8, CB2 and borated stainless steel grades ranging from low-alloy steel to high-alloy steel. Extended user-defined criteria and verified boundary conditions were used to predict the formation of A-segregations in cast steel. Both primary (PDAS) and secondary (SDAS) arm spacings were modelled as well. The optimized mould shape and the casting assembly were designed based on the simulation results.

**Keywords:** mould; iron-base alloy; steel; 34CrNiMo6; DHQ8; CB2; borated stainless steel; A-segregation; PDAS; SDAS



## 1. Introduction

The development of new iron-based alloys with strict requirements on the internal quality of a product is forcing manufacturers to optimize every step of the whole metallurgical process. A high degree of metallurgical cleanliness and homogeneous microstructure together with suitable hot working procedures lead to uniform mechanical properties in both the longitudinal and the transverse directions. At the beginning of all traditional metallurgical processes is the design of a mould. The mould geometry is always a compromise between axial shrinkage porosity and the chemical inhomogeneity of the alloy [1–4]. Ideally, from the point of view of internal quality, there should be a distinct mould geometry for different alloys. In the real world, such an approach cannot be cost efficient. Therefore, universal moulds, considering a company's portfolio, have been designed. During the 1Q/2021, the whole assembly will be manufactured and tested.

Currently used moulds in COMTES FHT Inc. are rather thin. A Height-to-Diameter ratio (H/D) reaches a critical value of 4.25 and a draft angle of 2°. Such geometry promotes a high chemical homogeneity. On the other hand, very high axial porosity is prominent as well. Poor internal quality is a limiting factor in subsequent initial stages of the hot working process, especially when considering highly alloyed experimental materials.

In this paper, all the relevant aspects of a mould design are considered. In addition to widely used approaches to design, both primary (PDAS) and secondary (SDAS) arm spacings and Suzuki criteria for A-segregation [5] are taken into account. These criteria are fundamental to subsequent material processing to achieve the desired final structure of the product. Moreover, various steel grades, with different material and metallurgical features,

are considered. To the best of the authors' knowledge, there is no previously published work presenting the medium-sized mould design in such a comprehensive manner.

## 2. Mould and Ingot Design

The mould geometry was designed in several logical steps. At first, the solidification curve was simulated in MAGMAsoft simulation software (version 5.4.1, MAGMASOFT, D-52072 Aachen, Germany) to try to understand the reason for the poor internal quality. CB2 steel (X12CrMoCoVNbN9-2-1) was used as the reference material. The set of simulations took into account different H/D ratios and different draft angles of the mould wall. Niyama criterion functions were used in two intervals, 0.3–1.3 and 0.3–2.6, respectively, hence verifying very strictly any potential areas of both microporosity and macroporosity [6,7]. Furthermore, criteria for determining primary dendrite arm spacing (PDAS) and secondary dendrite arm spacing (SDAS) were used. Based on these PDAS and SDAS values, a prior austenite grain size (PAGS) can be estimated.

The results are shown in Figure 1a,b and Figure 2a,b. Figure 1 shows the result of shrinkage prediction using the Niyama criterion (Ny) in the range of 0.3–1.3, H/D = 4.25 and draft angle 2° (upper left), draft angle 2.3° (upper right), H/D = 3.9 and draft angle 2.5° (lower left), H/D = 3.5 and draft angle 2.5° (lower right). It can be clearly seen that decreasing the H/D ratio and increasing the draft angle significantly enhances the internal quality.

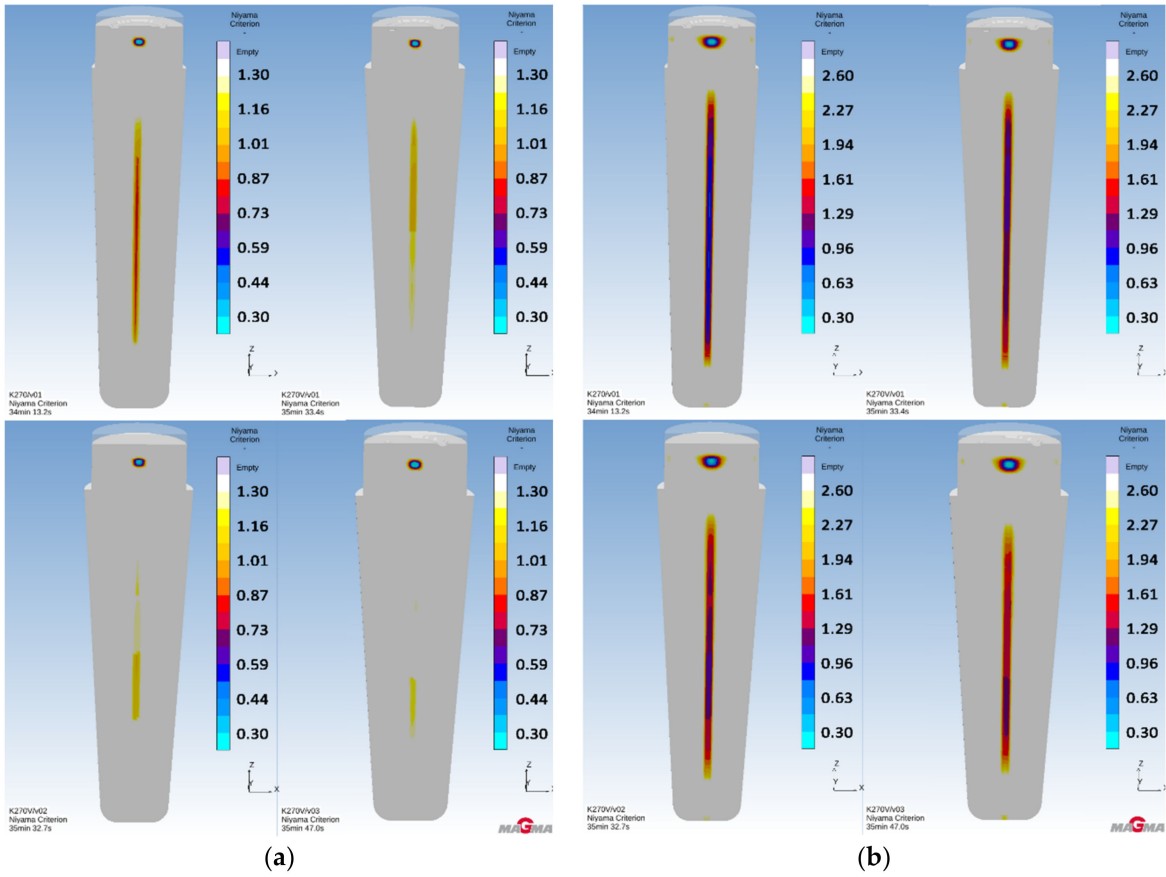

**Figure 1.** (**a**) Prediction of macroporosity—Ny 0.3–1.3, Height to Diameter (H/D) = 4.25, draft angle 2° (upper left), H/D = 4.25, draft angle 2.3° (upper right), H/D = 3.9, draft angle 2.5° (lower left), H/D = 3.5, draft angle 2.5° (lower right); (**b**) Prediction of microporosity—Ny 0.3–2.6, H/D = 4.25, draft angle 2° (upper left), H/D = 4.25, draft angle 2.3° (upper right), H/D = 3.9, draft angle 2.5° (lower left), H/D = 3.5, draft angle 2.5° (lower right).

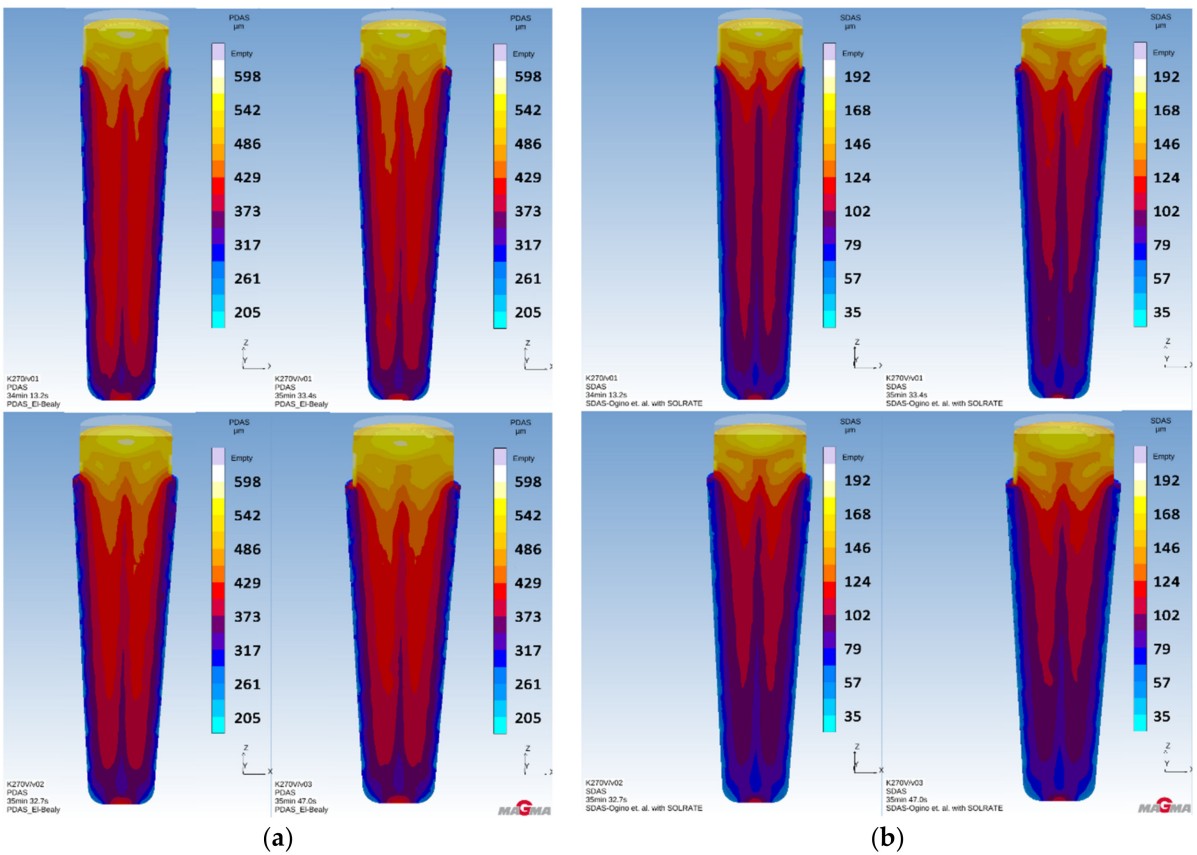

**Figure 2.** (**a**) Prediction of primary dendrite arm spacing (PDAS)—H/D = 4.25, draft angle 2° (202.1–594.5 μm)—(upper left), H/D = 4.25, draft angle 2.3° (202.3–599.4 μm)—(upper right), H/D = 3.9, draft angle 2.5° (205.0–598.5 μm)—(lower left), H/D = 3.5, draft angle 2.5° (204.5–600.7 μm)—(lower right); (**b**) Prediction of PDAS—H/D = 4.25, draft angle 2° (34.6–190.8 μm)—(upper left), H/D = 4.25, draft angle 2.3° (34.6–192.8 μm)—(upper right), H/D = 3.9, draft angle 2.5° (35.3–192.6 μm)—(lower left), H/D = 3.5, draft angle 2.5° (35.2–194.4 μm)—(lower right).

Figure 2a shows the result of PDAS prediction considering a mould with H/D = 4.25 and draft angle 2° (202.1–594.5 μm)—(upper left), H/D = 4.25 and draft angle 2.3° (202.3–599.4 μm)—(upper right), H/D = 3.9 and draft angle 2.5° (205.0–598.5 μm)—(lower left) and H/D = 3.5 and draft angle 2.5° (204.5–600.7 μm)—(lower right). Prior dendrite arm axes mildly increase with a decreasing H/D ratio and increasing draft angle.

Figure 2b shows the result of SDAS prediction considering a mould with H/D = 4.25 and draft angle 2° (34.6–190.8 μm)—(upper left), H/D = 4.25 and draft angle 2.3° (34.6–192.8 μm)—(upper right), H/D = 3.9 and draft angle 2.5° (35.3–192.6 μm)—(lower left) and H/D = 3.5 and draft angle 2.5° (35.2–194.4 μm)—(lower right). Secondary dendrite arm axes mildly increase with a decreasing H/D ratio and increasing draft angle.

The results involving the Niyama criterion for microporosity and macroporosity show gradually increasing internal quality with respect to the length of the ingot taper. However, the beneficial effect for subsequent hot working is marginal. The results of PDAS and SDAS prediction show increasing spacing along with a decreasing H/D ratio and increasing draft angle, which negatively affects the prior austenite grain coarsening.

The apparently unsatisfactory material features led to a new mould design which tried to diminish the axial porosity together with a low degree of cross-sectional chemical inhomogeneity. For the purposes of the new mould design, the following criteria for the ingot body (Figure 3) were defined: $D_{av}$ 300–350 mm, L 550–600 mm, head volume 17–20%, total ingot mass 450–500 kg, the flexible wall thickness ranging from 70 mm at the top to 120 mm at the bottom of the mould, and the wall thickness of 300 mm for the lower part of the teeming set.

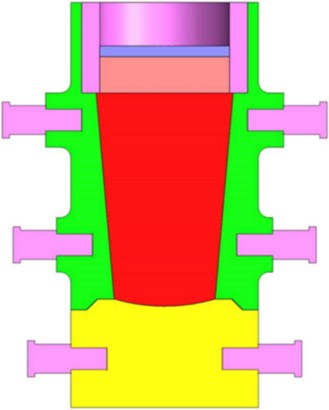 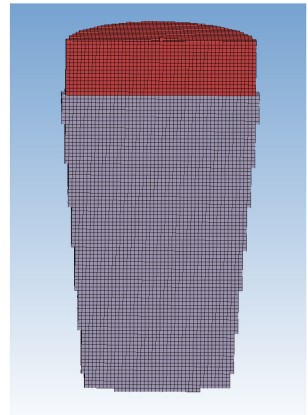

**Figure 3.** Cross-section view of mould assembly (**left**); Mesh consisting of 169,629 cubic unit cells (7 mm × 7 mm × 7 mm) (**right**).

Both the top teeming and the bottom teeming arrangements were considered. The simulations of filling and solidifying processes including a wide range of materials (low-alloy, mid-alloy and high alloy steel grades) were performed. For the purposes of this paper, only top teeming is discussed in detail.

### 2.1. Numerical Model

Simulation of the teeming cycle, solidification and the cooling of the ingot were performed in MAGMAsoft software [8]. The simulation of filling and solidifying is as follows: the velocity field of the moving fluid during the filling is calculated in accordance with Newton's second law (Equations (1) and (2)) and the mass continuity equation (Equation (3)) assuming Newtonian incompressible fluid only. The mathematical description of this is [9]:

$$\sigma_{ji,j} + p_i = \rho u_i + \rho(u_j u_{i,j}), \tag{1}$$

$$\sigma_{ij} = -\delta_{ij}p + \mu(u_{i,j} + u_{j,i}), \tag{2}$$

$$u_{i,i} = 0, \tag{3}$$

where $\sigma$ represents the component of external stress (normal stress or shear stress), $p$ is the pressure, $\rho$ is the density of a fluid and $u$ is the velocity vector. This mathematical description is in accordance with the Law of Conservation of Mass including those terms describing the flow of molten metal needed for the calculation of the thermal fields during the filling (Equation (4)) [10].

$$\rho c_p \dot{T} + u_i(\rho c_p T)_{,i} = (kT_{,i})_{,i} + \dot{Q}''', \tag{4}$$

where $\rho$ is the density, $c_p$ is the specific heat, $\dot{T}$ is the temperature difference, $u$ is the velocity vector, $k$ is the thermal conductivity and $\dot{Q}'''$ is the internal heat generation term. The subsequent solidifying analysis is based on the Law of Conservation of Energy (Equation (4)) as well, in combination with the release of latent heat. It must be emphasized that during the solidification the second term on the left-hand side represents the natural thermal convection in the molten metal and the two-phase region. From this, it is clear that the thermal convection can greatly affect the thermal field during solidification.

The partial differential equations in the numerical analysis (Equations (1)–(4)) were solved and linked together for both the filling and the subsequent solidification processes. The numerical model for predicting shrinkage and microsegregation is discussed in detail in the literature [11–16], explaining the equations for simulating multicomponent peritectic solidification.

The new mould design was evaluated with respect to the Niyama criterion, Gradient Time, A-segregation and user-defined criteria for primary dendrite arm spacing (PDAS)

and secondary dendrite arm spacing (SDAS). Porosity evaluation is represented by the Niyama criterion (Equation (5)) [16–18] that relates the origin and the location of thermal-axis porosity to the thermal gradient G (K/mm) and the cooling rate (K/s). The thermal axis of an ingot generally represents an area with low thermal gradient where the solid–liquid volumetric contraction cannot be sufficiently compensated by the molten metal filled from the head part of the ingot. Due to the low thermal gradient there is a lack of driving force needed for pushing the molten metal through the solid–liquid mixture, the so called the 'mushy zone' [10]. MAGMAsoft simulation software uses the Niyama criterion defined in $K^{1/2} s^{1/2} mm^{-1}$ units. For conversion to conventional Niyama criterion units, the following relation applies: $1 K^{1/2} min^{1/2} cm^{-1}$ equals $0.775 K^{1/2} s^{1/2} mm^{-1}$.

$$\text{Niyama} = G/\sqrt{R} \tag{5}$$

The Gradient Time criterion depends on the progress of solidification. For a given time, one can calculate the local thermal transitions in an ingot. The behavior of thermal gradients may be useful for predicting difficulties in compensating volumetric changes during solidification.

A-segregates (Suzuki Criterion) (Equation (6)) arise due to the flow of solute-rich interdendritic fluid via thermosolutal convection. They are characterized in the final solidified microstructure as channels of enriched solid, which can have a near-eutectic composition. Their formation mechanism can be described as follows: in steels, the enriched inter-dendritic liquid will often be less dense than the bulk liquid (due to enrichment in light elements such as C, Mn, Si, S and P), and will hence tend to rise. As the liquid moves towards the bulk liquid and the top of the ingot, it will increase in temperature, but its composition will remain nearly constant due to slow mass diffusion (it will therefore be out of equilibrium with the solid it encounters). This hotter enriched liquid then causes delayed growth or remelting of the solid around it, creating persistent solute-enriched channels. A-segregates are also commonly referred to as 'channel segregates', or 'freckles' or 'chimneys' when they arise in directionally solidified ingots.

$$R^{1.1} \dot{T} = S, \tag{6}$$

where R is the solidification (or isotherm) speed in $\mu m\ s^{-1}$, $\dot{T}$ is the cooling rate in $K\ s^{-1}$, and S is termed here the Suzuki number in $K\ s^{-1}\ (\mu m\ s^{-1})^{1.1}$ [15,17,19–21].

The PDAS (Primary Dendrite Arm Spacing) criterion ($\mu m$) (Equation (7)) [15,22] shows the gap between the primary arms of the dendrites in micrometres. This criterion, together with the SDAS criterion, gives an idea of the size of the primary austenitic grain after solidification of the ingot.

$$\text{El-Bealy: } \lambda_1 = 279\ \dot{T}^{-0.206}C^{-0.019-0.492C}{}_c \tag{7}$$

$\lambda_1$ expression only valid for $0.15 < C_C < 1.0$

The SDAS (Secondary Dendrite Arm Spacing) criterion ($\mu m$) (Equation (8)) [15,23] shows the gap between the secondary arms of the dendrites in micrometres. This criterion, together with the PDAS criterion, gives an idea of the size of the primary austenitic grain after solidification of the ingot.

$$\text{Ogino et al.: } \lambda_2 = 123\ \dot{T}^{-0.33}e^{(-0.281C_c+0.175C_{Mn}-0.063C_{Cr}-00136C_{Mo}-0.091C_{Ni})} \tag{8}$$

### 2.2. Numerical Simulation Settings

The numerical computational mesh for the ingot (457–483 kg) consisted of 169,629 controlling elements for a high precision calculation (Figure 3). The teeming set consisted of 1,465,000 controlling elements.

The chemical compositions of the individual materials are shown in Table 1; CB2 (X12CrMoCoVNbN9-2-1)—COST 536, ATABOR (X1CrNiBMo19-13-2-1)—DIN 1.4696, ASTM

A887M type 304B6, DHQ8 (8Cr4SiMo)—ASTM A681 type A2 modif. and 34CrNiMo6—EN10083-3. The mould is made of hematite cast iron with lamellar graphite. Temperatures of the melt and the teeming set components before the filling are shown in Table 2. Temperature-dependent thermo-physical properties of all the tested materials were extracted from a database integrated in computational software (Table 3). Other temperature-dependent parameters like the density, the coefficient of linear thermal expansion, the specific enthalpy, the fraction solid, the viscosity, the permeability and the Poisson's ratio were generated from JMatPro simulation software.

**Table 1.** A list of materials used for the simulation. Chemical composition in wt.%.

| Grade | C | Si | Mn | P | S | Cr | Ni | Mo | Co | V | B |
|--------|-------|------|------|-------|-------|------|-------|------|-------|-------|--------|
| CB2 | 0.11 | 0.36 | 0.84 | 0.008 | 0.004 | 9.20 | 0.16 | 1.43 | 0.98 | 0.22 | 0.007 |
| ATABOR | 0.014 | 0.46 | 1.24 | 0.027 | 0.001 | 19.00 | 12.80 | 0.93 | 0.06 | 0.05 | 1.55 |
| DHQ8 | 0.76 | 0.74 | 0.58 | 0.007 | 0.001 | 4.00 | 0.12 | 0.55 | 0.012 | 0.043 | 0.0003 |
| 34CrNiMo6 | 0.34 | 0.10 | 0.80 | 0.007 | 0.001 | 1.56 | 1.56 | 0.25 | 0.011 | 0.03 | 0.0001 |
| Cast iron | 3.05 | 0.60 | 1.75 | 0.030 | 0.030 | - | - | - | - | - | - |

**Table 2.** Temperatures of the melt and the teeming set components before the filling.

| Grade | Initial Temperature (°C) | Mould (°C) | Exo Sleeve (°C) | Hot Topping (°C) |
|--------|--------------------------|------------|-----------------|------------------|
| CB2 | 1570 | 40 | 20 | 20 |
| ATABOR | 1390 | 40 | 20 | 20 |
| DHQ8 | 1540 | 40 | 20 | 20 |
| 34CrNiMo6 | 1570 | 40 | 20 | 20 |

**Table 3.** Main thermodynamic parameters.

| Grade | Tsolid (°C) | Tliquid (°C) | Latent Heat (kJ/kg) |
|--------|-------------|--------------|---------------------|
| CB2 | 1408 | 1493 | 206 |
| ATABOR | 1206 | 1261 | 206 |
| DHQ8 | 1317 | 1447 | 221 |
| 34CrNiMo6 | 1432 | 1491 | 237 |

## 3. Results and Discussion

Evaluation of the shape of the shrinkage and head volume sufficiency is based on the 'Soundness' criterion. The 'Soundness' criterion allows you to display the local percentage of metal at the end of the solidification simulation and thus to determine the quality of the feeding of the casting. One hundred percent for 'Soundness' corresponds to 0 percent for 'Porosity' and vice versa. The Gradient Time criterion (°C/mm) helps to evaluate the progress of the solidification in time, thus explaining the origins of the axial macroporosity and microporosity. The total amount of ingot inhomogeneity is represented by the Total Porosity criterion. Shrinkage porosity is evaluated by the Niyama criterion within the range of 0.3–1.3 $K^{1/2}s^{1/2}$ $mm^{-1}$. Microporosity is evaluated within the range of 0.3–2.6 $K^{1/2}$ $s^{1/2}$ $mm^{-1}$.

The Suzuki criterion ($K s^{-1}(\mu m s^{-1})^{1.1}$) was used to evaluate A-segregation. White regions on the figures below show the as-cast structure with a low amount of, or even without, A-segregation. For a given alloy, the value of a given criterion is higher than its critical value. Values below the critical value indicate higher probability of A-segregation. The lower the value, the coarser and broader the segregation. More pronounced segregation is generated towards the ingot's head. Both the amount and the position of the segregation of carbon, silicon, chromium, nickel and molybdenum is evaluated for all the steel grades mentioned above. Primary dendrite arm spacing (PDAS) and secondary dendrite arm spacing (SDAS) criteria were also taken into account. However, with high-alloy steel grades (CB2 and ATABOR), the PDAS results are rather approximate due to the $C_C$ value being out of the validity region $0.15 < C_C < 1.0$ (Equation (7)).

### 3.1. Solidification of CB2 Steel Ingot

The CB2 ingot simulation shows that the head volume is sufficient, since the shrinkage did not affect the ingot's body and the head still possesses a surplus of sound material (Figure 4a). The temperature gradient during the course of the solidification shows that 26 min 42s after the filling, the temperature field is separated (Figure 4b). The temperature field separation means we can expect the occurrence of microporosity. On the other hand, macroporosity will not arise. The Total Porosity criterion shows the total amount of inhomogeneities that can be expected (Figure 4c). Regions of axial microporosity and surface shrinkage can be clearly seen. The Niyama criterion 0.3–1.3 $K^{1/2}$ $s^{1/2}$ $mm^{-1}$ shows no macroporosity (Figure 4d). However, the Niyama criterion 0.3–2.6 $K^{1/2}$ $s^{1/2}$ $mm^{-1}$ reveals a region with microporosity (Figure 4e). In Figure 4f, the Suzuki criterion 0.0–0.015 $K$ $s^{-1}(\mu m$ $s^{-1})^{1.1}$ is shown (A-segregation).

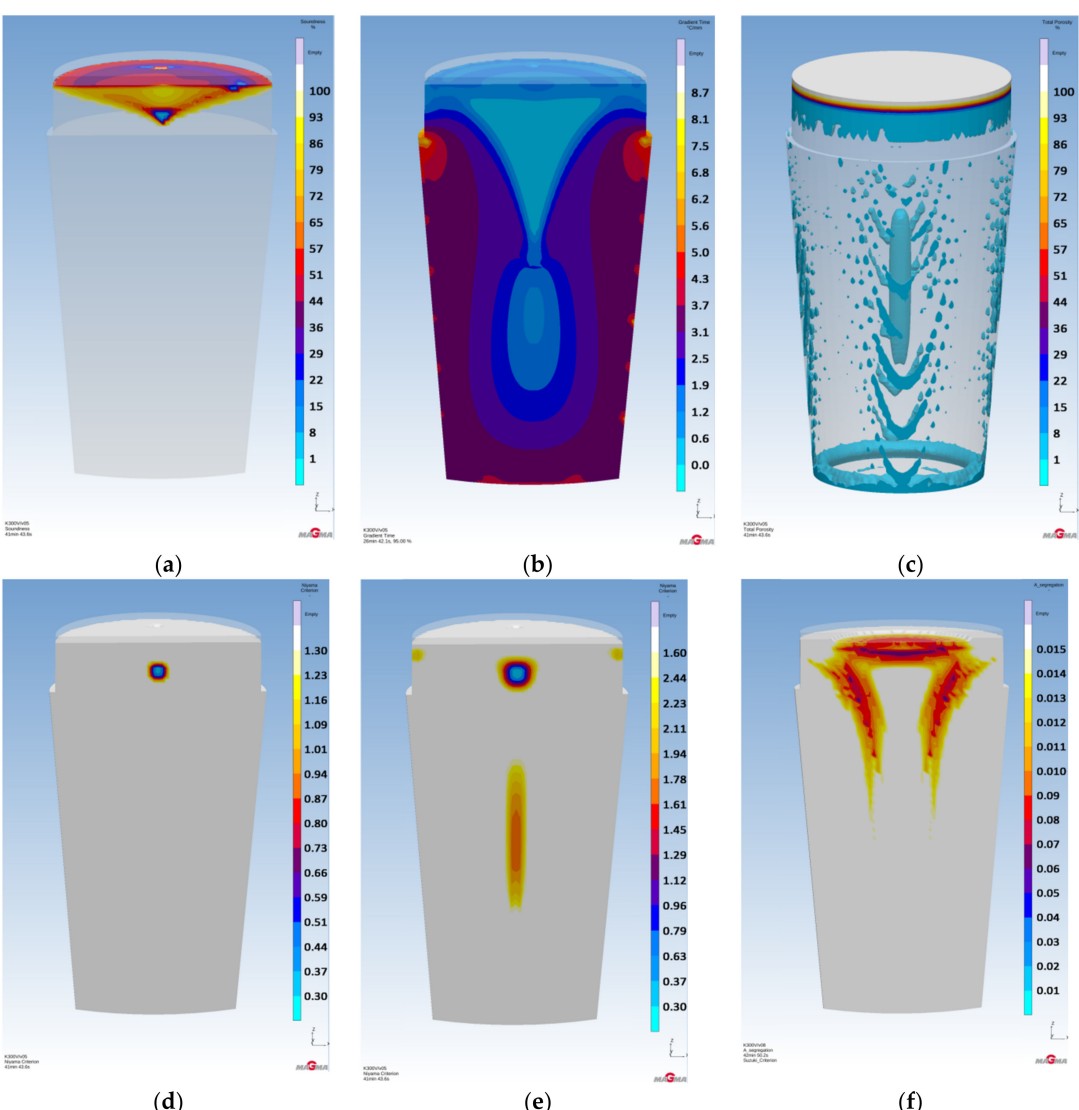

**Figure 4.** The results of criteria: (**a**) Soundness %; (**b**) Gradient Time 26 min 42.1 s; (**c**) Total Porosity %; (**d**) Niyama 0.3–1.3 $K^{1/2}$ $s^{1/2}$ $mm^{-1}$; (**e**) Niyama 0.3–2.6 $K^{1/2}$ $s^{1/2}$ $mm^{-1}$; (**f**) A-segregation (Suzuki criterion) 0.0–0.015 $K$ $s^{-1}(\mu m$ $s^{-1})^{1.1}$.

Element Segregation, PDAS and SDAS in CB2 Steel Ingot

The results of the segregation prediction of CB2 steel ingot indicate that the concentration of carbon (Figure 5a) varies from 0.03 to 0.19 wt.% assuming the melt analysis value 0.11 wt.%, the concentration of silicon (Figure 5b) varies from 0.28 to 0.44 wt.% assuming

the melt analysis value 0.36 wt.%, the concentration of chromium (Figure 5c) varies from 8.68 to 9.72 wt.% assuming the melt analysis value 9.20 wt.%, the concentration of nickel (Figure 5d) varies from 0.14 to 0.18 wt.% assuming the melt analysis value 0.16 wt.%, and the concentration of molybdenum (Figure 5e) varies from 1.19 to 1.64 wt.% assuming the melt analysis value 1.43 wt.%. Figure 5f shows the prediction of primary dendrite arm spacing (PDAS) varying from 203.3 to 605.0 μm. Figure 5g shows the prediction of secondary dendrite arm spacing (SDAS) varying from 34.9 to 198.6 μm.

### 3.2. Solidification of ATABOR Steel Ingot

The ATABOR ingot simulation shows that the head volume is sufficient, since the shrinkage did not affect the ingot's body and the head still possesses a surplus of sound material (Figure 6a). The temperature gradient during the course of the solidification shows that 25 min 4s after the filling, the temperature field is separated (Figure 6b). The temperature field separation means microporosity can be expected. On the other hand, macroporosity will not arise. The Total Porosity criterion shows the total amount of inhomogeneities that can be expected (Figure 6c). Regions of axial microporosity and surface shrinkage can be clearly seen. The Niyama criterion 0.3–1.3 $K^{1/2}$ $s^{1/2}$ $mm^{-1}$ shows no macroporosity (Figure 6d). However, the Niyama criterion 0.3–2.6 $K^{1/2}$ $s^{1/2}$ $mm^{-1}$ shows a region with microporosity (Figure 6e). Figure 6f shows the Suzuki criterion 0.0–0.019 $K$ $s^{-1}(\mu m$ $s^{-1})^{1.1}$ (A-segregation).

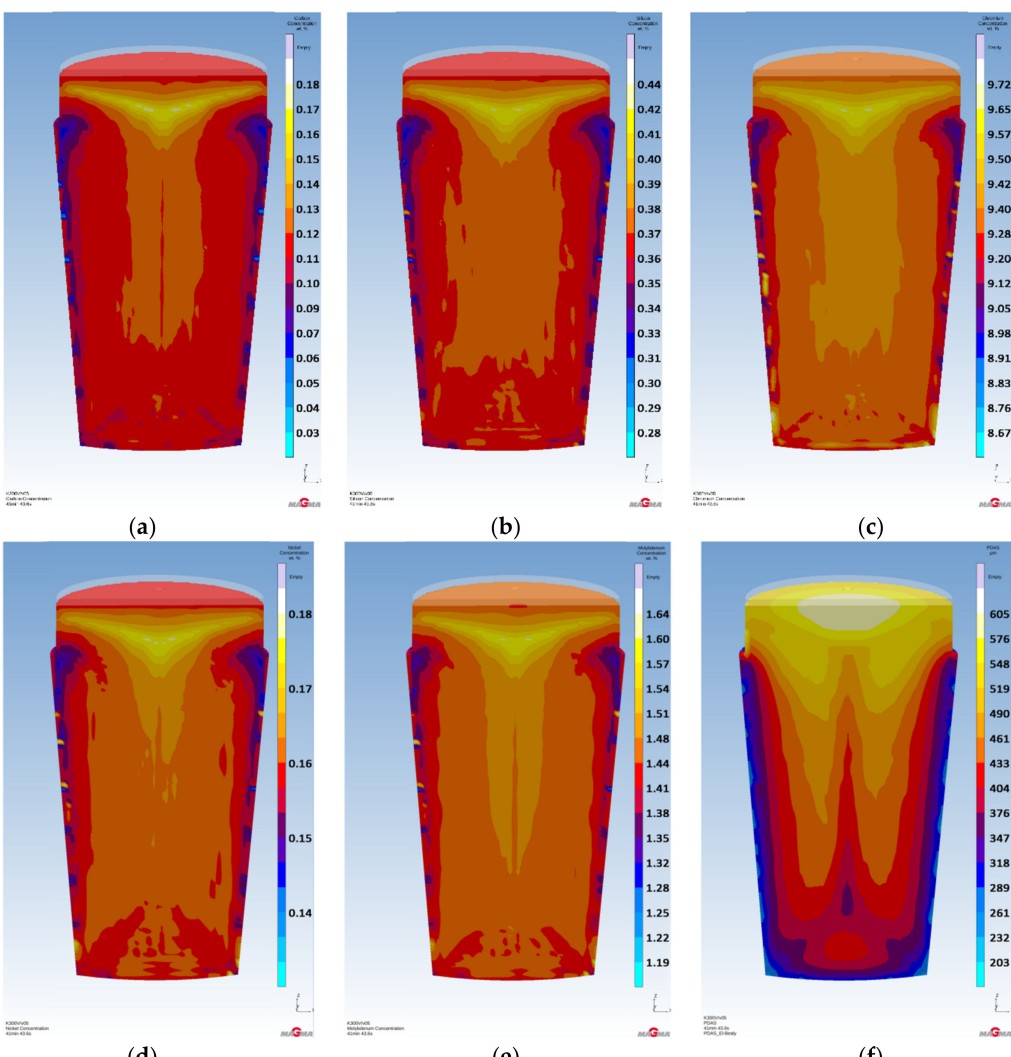

**Figure 5.** *Cont..*

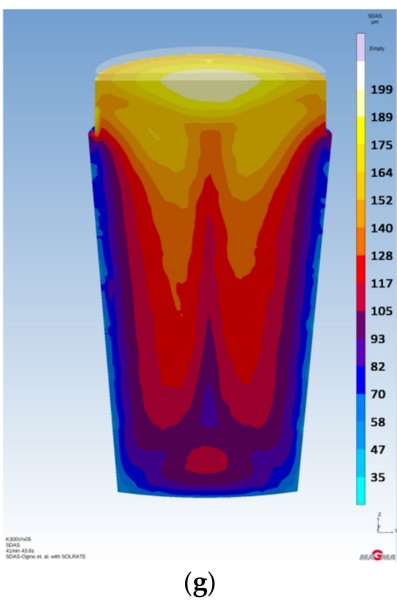

(**g**)

**Figure 5.** Element segregation, PDAS and secondary dendrite arm spacing (SDAS): (**a**) [wt.% C]; (**b**) [wt.% Si]; (**c**) [wt.% Cr]; (**d**) [wt.% Ni]; (**e**) [wt.% Mo]; (**f**) PDAS; (**g**) SDAS.

Element Segregation, PDAS and SDAS in ATABOR Steel Ingot

The segregation prediction of ATABOR steel ingot indicates that the concentration of carbon (Figure 7a) varies from 0.006 to 0.020 wt.% assuming the melt analysis value 0.014 wt.%, the concentration of silicon (Figure 7b) varies from 0.33 to 0.57 wt.% assuming the melt analysis value 0.46 wt.%, the concentration of chromium (Figure 7c) varies from 18.22 to 20.05 wt.% assuming the melt analysis value 19.00 wt.%, the concentration of nickel (Figure 7d) varies from 11.95 to 13.63 wt.% assuming the melt analysis value 12.80 wt.%, and the concentration of molybdenum (Figure 7e) varies from 0.77 to 1.08 wt.% assuming the melt analysis value 0.93 wt.%. Figure 7f shows the prediction of primary dendrite arm spacing (PDAS) varying from 193.1 to 557.8 μm. Figure 7g shows the prediction of secondary dendrite arm spacing (SDAS) varying from 2.18 to 34.78 μm.

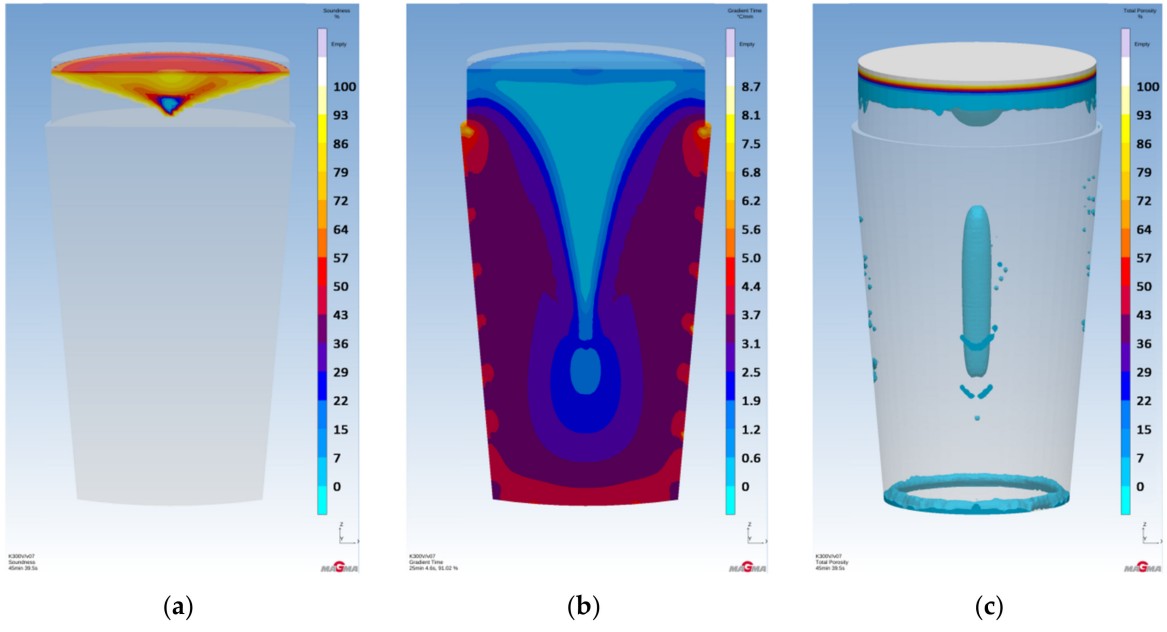

(**a**)　　　　　　　　　　　　(**b**)　　　　　　　　　　　　(**c**)

**Figure 6.** *Cont.*

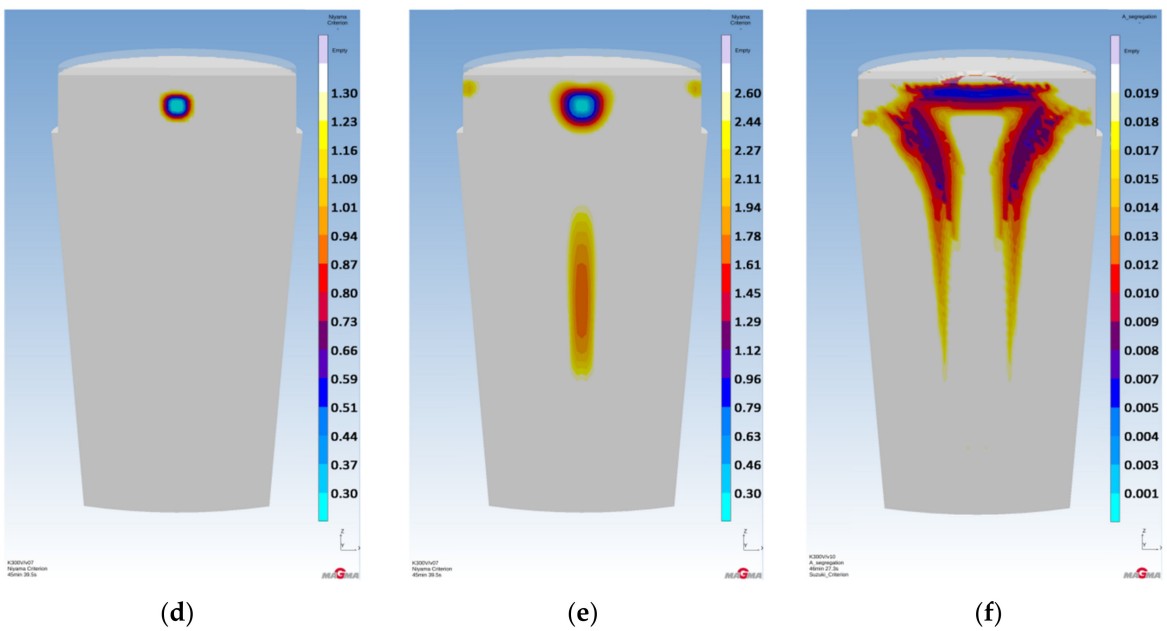

### 3.3. Solidification of DHQ8 Steel Ingot

The DHQ8 steel ingot simulation indicate that the head volume is sufficient, since the shrinkage did not affect the ingot's body. However, the head possesses very little surplus of sound material (Figure 8a). The temperature gradient during the course of the solidification shows that 31 min 24 s after the filling, the temperature field is separated (Figure 8b). The temperature field separation means the occurrence of microporosity can be expected. On the other hand, macroporosity wil not arise. The Total Porosity criterion shows the total amount of inhomogeneities that can be expected (Figure 8c). Regions of axial microporosity can be clearly seen. The surface shrinkage is almost negligible. The Niyama criterion 0.3–1.3 $K^{1/2}$ $s^{1/2}$ $mm^{-1}$ shows no macroporosity (Figure 8d). However, the Niyama criterion 0.3–2.6 $K^{1/2}$ $s^{1/2}$ $mm^{-1}$ reveals a region with microporosity (Figure 8e). Figure 8f shows the Suzuki criterion 0.0–0.03 $K$ $s^{-1}(\mu m\ s^{-1})^{1.1}$ (A-segregation).

Element Segregation, PDAS and SDAS DHQ8 Steel Ingot

The segregation prediction of DHQ8 steel ingot indicates that the concentration of carbon (Figure 9a) varies from 0.42 to 1.14 wt.% assuming the melt analysis value 0.76 wt.%, the concentration of silicon (Figure 9b) varies from 0.63 to 0.89 wt.% assuming the melt analysis value 0.74 wt.%, the concentration of chromium (Figure 9c) varies from 3.32 to 4.77 wt.% assuming the melt analysis value 4.00 wt.%, the concentration of nickel (Figure 9d) varies from 0.12 to 0.13 wt.% assuming the melt analysis value 0.12 wt.%, and the concentration of molybdenum (Figure 9e) varies from 0.36 to 0.76 wt.% assuming the melt analysis value 0.55 wt.%. Figure 9f shows the prediction of primary dendrite arm spacing (PDAS) varying from 208.5 to 545.9 μm. Figure 9g shows the prediction of secondary dendrite arm spacing (SDAS) varying from 44.1 to 207.6 μm.

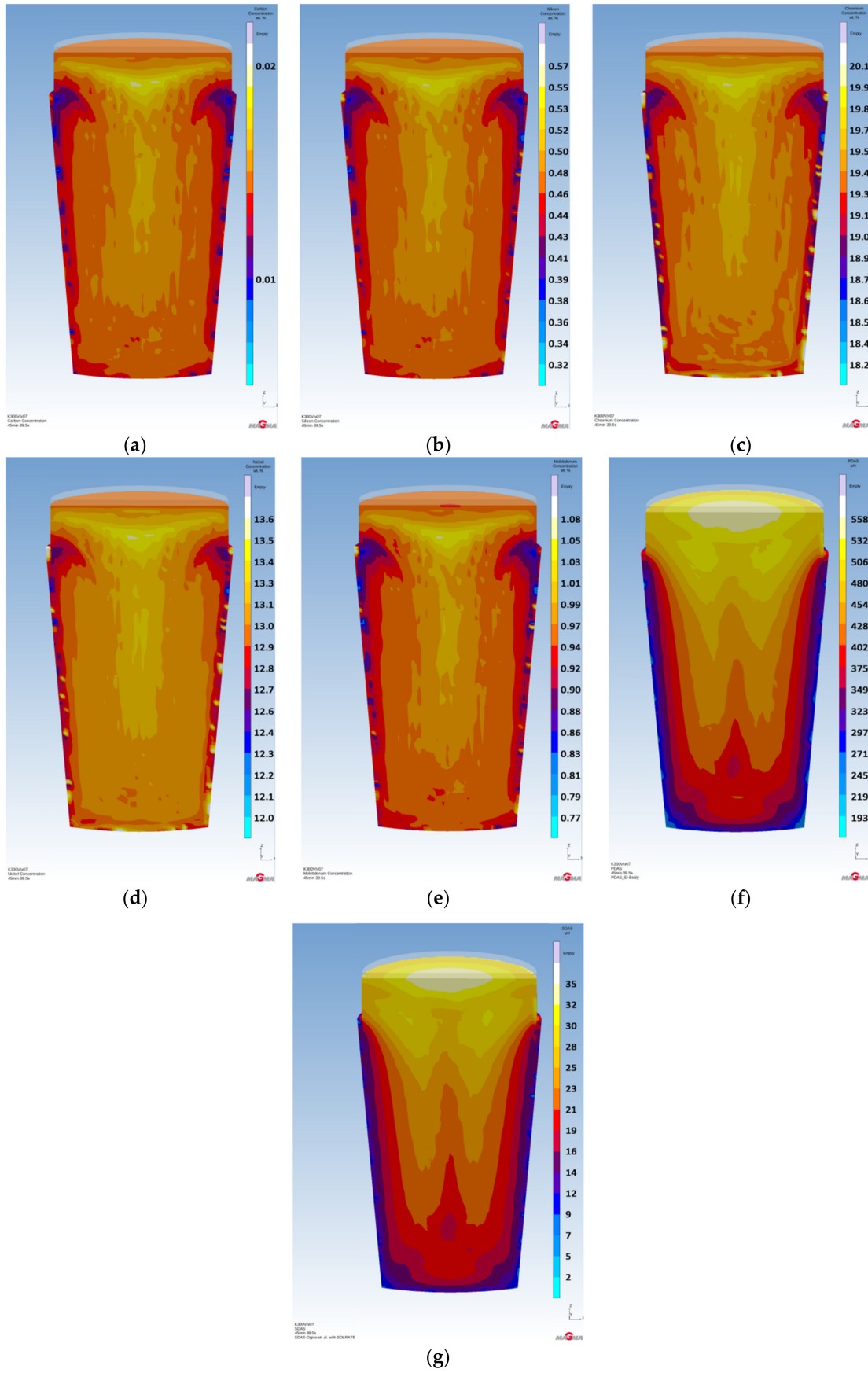

**Figure 7.** Element segregation, PDAS and SDAS: (**a**) [wt.% C]; (**b**) [wt.% Si]; (**c**) [wt.% Cr]; (**d**) [wt.% Ni]; (**e**) [wt.% Mo]; (**f**) PDAS; (**g**) SDAS.

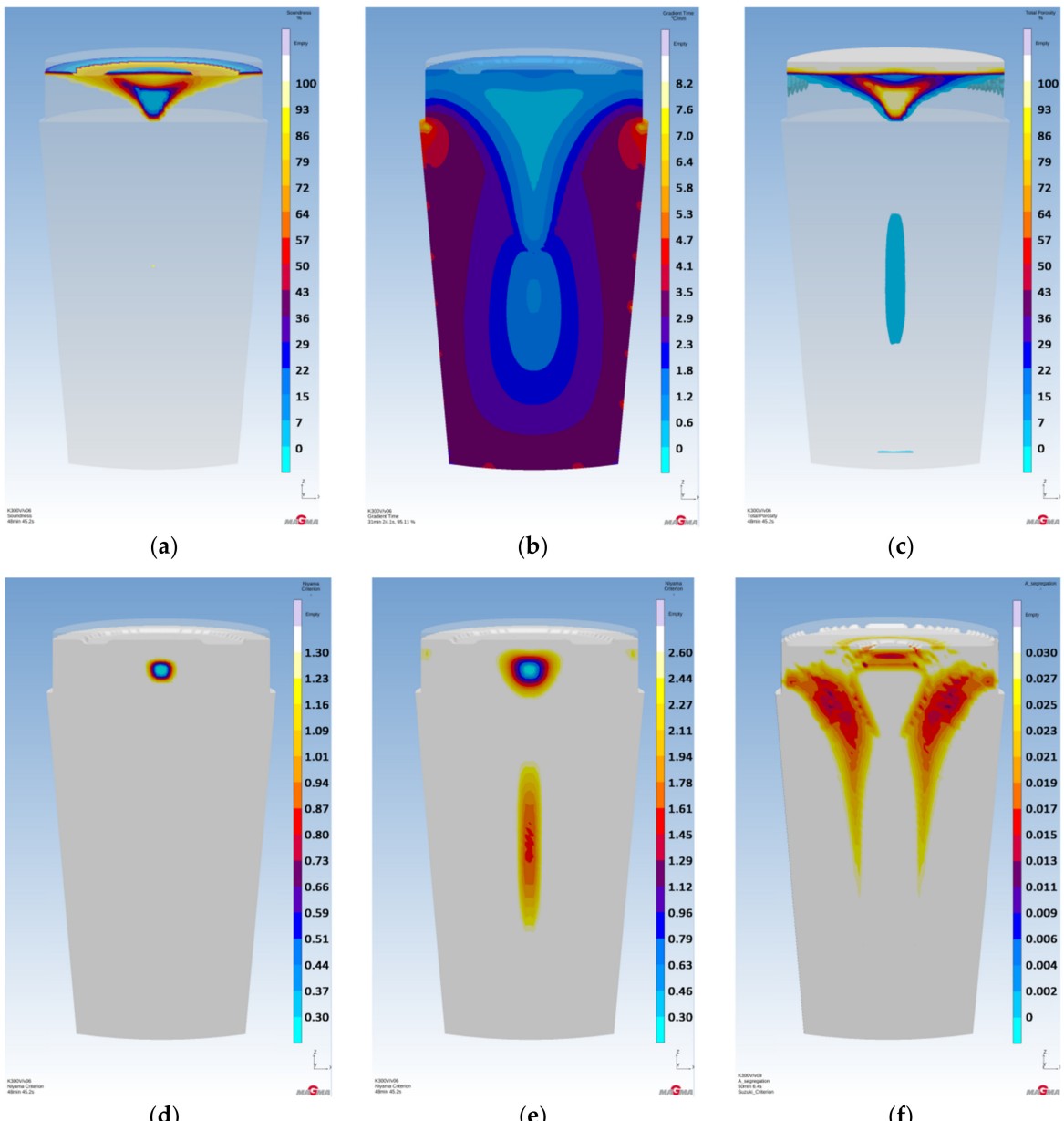

**Figure 8.** The results of criteria: (**a**) Soundness %; (**b**) Gradient Time 31 min 24.1s; (**c**) Total Porosity %; (**d**) Niyama 0.3–1.3 K$^{1/2}$ s$^{1/2}$ mm$^{-1}$; (**e**) Niyama 0.3–2.6 K$^{1/2}$ s$^{1/2}$ mm$^{-1}$; (**f**) A-segregation (Suzuki criterion) 0.0–0.03 K s$^{-1}$(μm s$^{-1}$)$^{1.1}$.

### 3.4. Solidification of 34CrNiMo6 Steel Ingot

The 34CrNiMo6 ingot simulation indicates that the head volume is sufficient, since the shrinkage did not affect the ingot's body and the head still possesses a surplus of sound material (Figure 10a). The temperature gradient during the course of the solidification shows that 27 min 13 s after the filling, the temperature field is separated (Figure 10b). The temperature field separation means microporosity can be expected. On the other hand, macroporosity wil not arise. The Total Porosity criterion shows the total amount of inhomogeneities that can be expected (Figure 10c). Small regions of axial microporosity and surface shrinkage can be clearly seen. The Niyama criterion 0.3–1.3 K$^{1/2}$ s$^{1/2}$ mm$^{-1}$ shows no macroporosity (Figure 10d). However, the Niyama criterion 0.3–2.6 K$^{1/2}$ s$^{1/2}$ mm$^{-1}$ shows a region with microporosity (Figure 10e). Figure 10f shows the Suzuki criterion 0.0–0.010 K s$^{-1}$(μm s$^{-1}$)$^{1.1}$ (A-segregation).

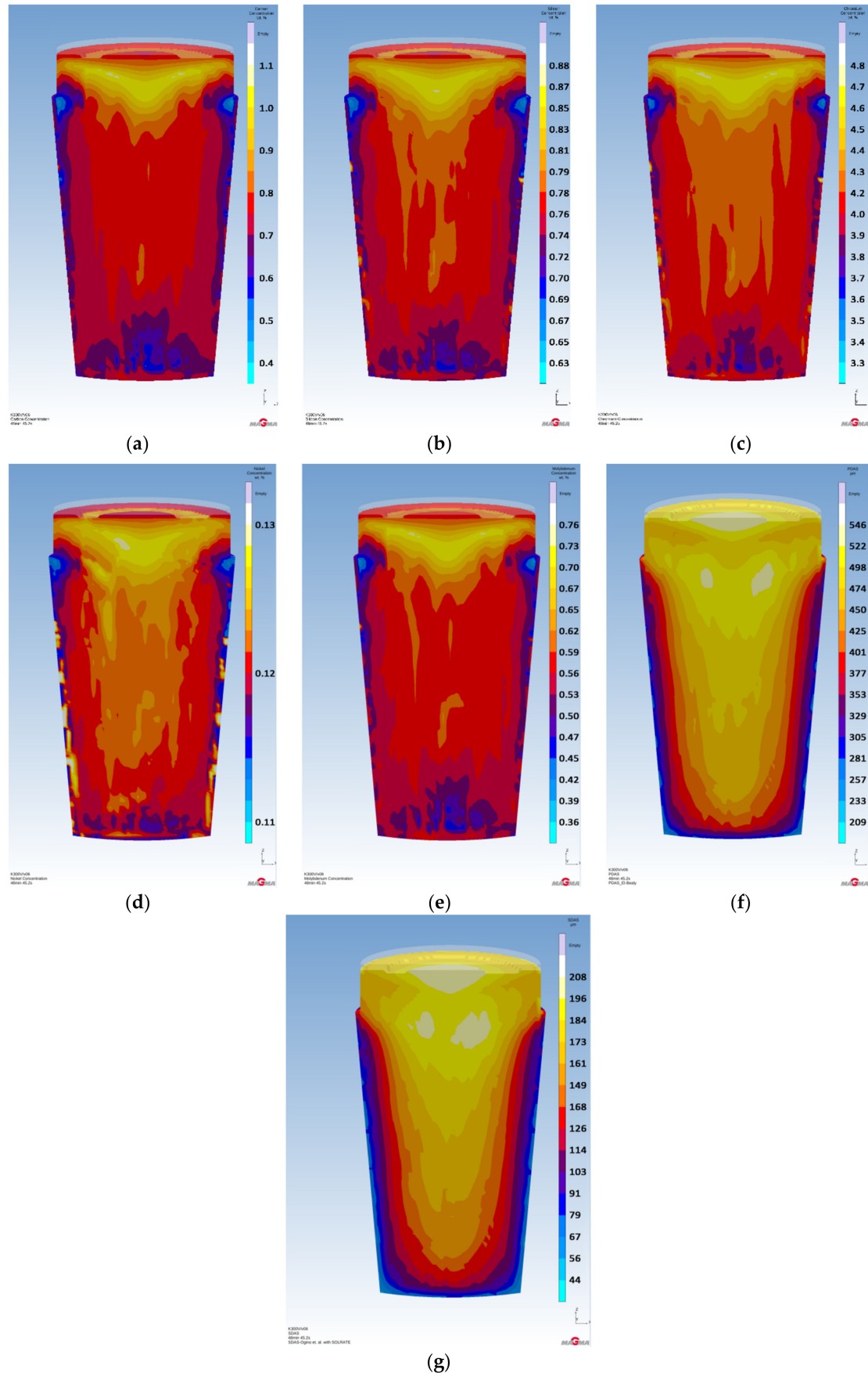

**Figure 9.** Element segregation, PDAS and SDAS: (**a**) [wt.% C]; (**b**) [wt.% Si]; (**c**) [wt.% Cr]; (**d**) [wt.% Ni]; (**e**) [wt.% Mo]; (**f**) PDAS; (**g**) SDAS.

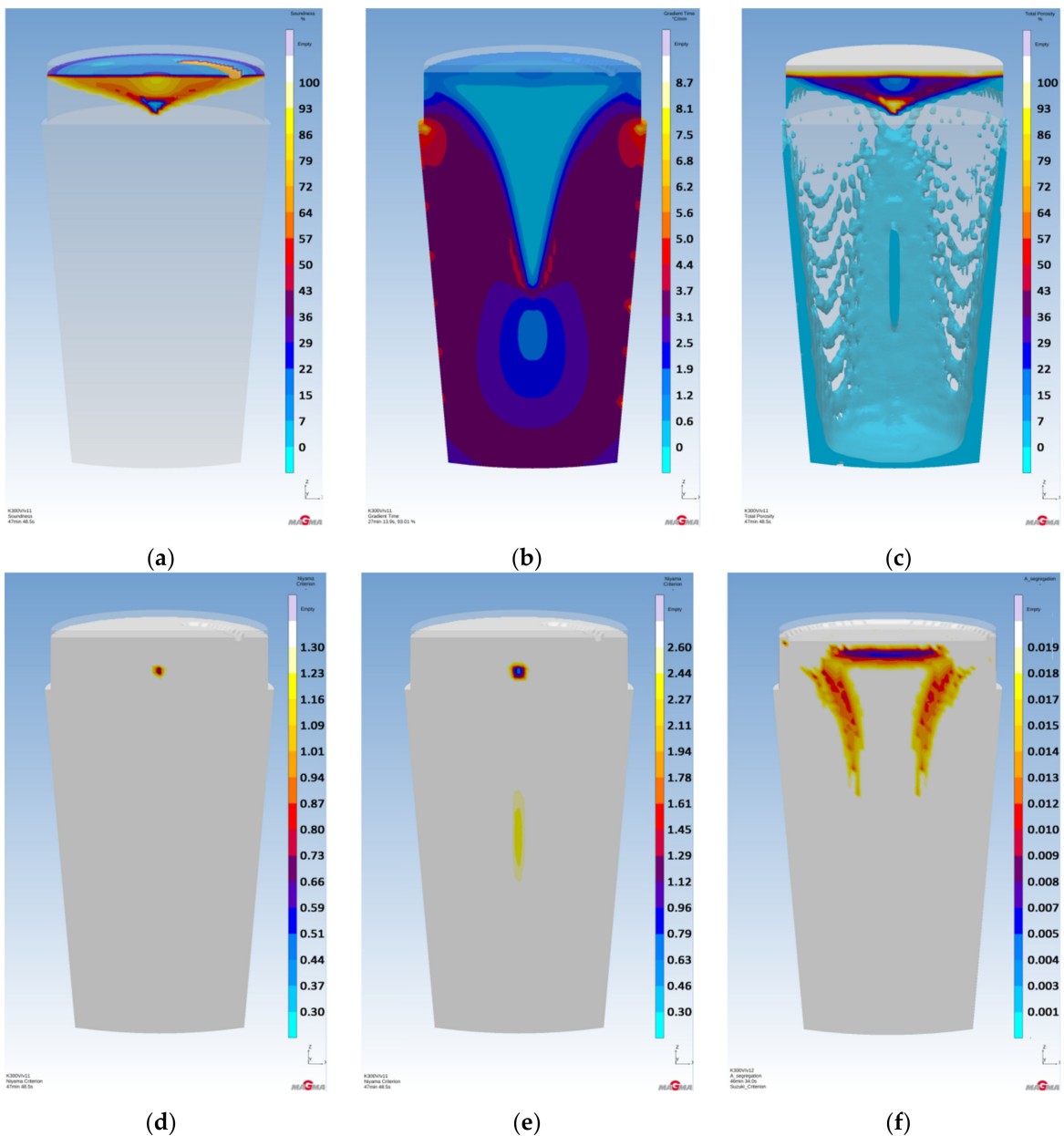

**Figure 10.** The results of criteria: (**a**) Soundness %; (**b**) Gradient Time 27 min 13.9 s; (**c**) Total Porosity %; (**d**) Niyama 0.3–1.3 $K^{1/2} s^{1/2} mm^{-1}$; (**e**) Niyama 0.3–2.6 $K^{1/2} s^{1/2} mm^{-1}$; (**f**) A-segregation (Suzuki criterion) 0.0–0.010 $K s^{-1}(\mu m s^{-1})^{1.1}$.

Element Segregation, PDAS and SDAS in 34CrNiMo6 Steel Ingot

The segregation prediction of 34CrNiMo6 steel ingot indicates that the concentration of carbon (Figure 11a) varies from 0.15 to 0.62 wt.% assuming the melt analysis value 0.34 wt.%, the concentration of silicon (Figure 11b) varies from 0.07 to 0.14 wt.% assuming the melt analysis value 0.10 wt.%, the concentration of chromium (Figure 11c) varies from 1.44 to 1.72 wt.% assuming the melt analysis value 1.56 wt.%, the concentration of nickel (Figure 11d) varies from 1.52 to 1.64 wt.% assuming the melt analysis value 1.56 wt.%, and the concentration of molybdenum (Figure 11e) varies from 0.16 to 0.37 wt.% assuming the melt analysis value 0.25 wt.%. Figure 11f shows the prediction of primary dendrite arm spacing (PDAS) varying from 226.8 to 683.2 μm. Figure 11g shows the prediction of secondary dendrite arm spacing (SDAS) varying from 52.5 to 305.2 μm.

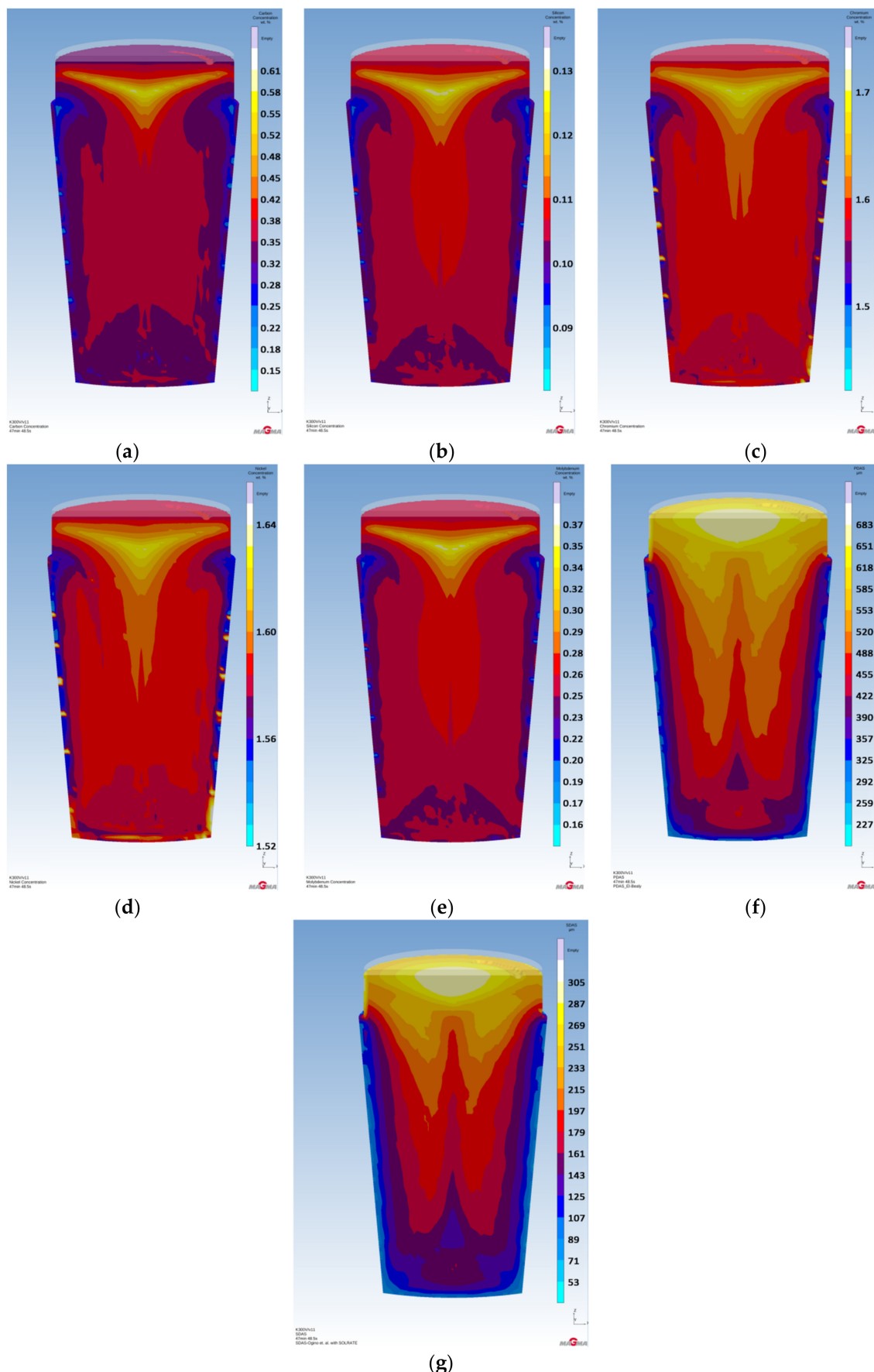

**Figure 11.** Element segregation, PDAS and SDAS: (**a**) [wt.% C]; (**b**) [wt.% Si]; (**c**) [wt.% Cr]; (**d**) [wt.% Ni]; (**e**) [wt.% Mo]; (**f**) PDAS; (**g**) SDAS.

## 4. Conclusions

For the 500 kg conical steel ingot, an optimized value of the Height-to-Diameter ratio (H/D = 1.65) was proposed. Analysis of the results from the numerical simulations in Section 3 shows a significant improvement of the optimized mould and ingot geometry in comparison to the currently used geometry shown in the Section 1—specifically, the well-balanced compromise between the microporosity and segregation for all given steel grades.

The assessment of various mould and ingot geometries is based on the prediction of both the macroporosity and the microporosity using the Niyama criterion, A-segregation using the Suzuki criterion and also primary dendrite arm spacing (PDAS) and secondary dendrite arm spacing (SDAS). Both PDAS and SDAS parameters are closely related to the solidification time and the chemical inhomogeneity.

Assuming the round shape of the mould and the ingot, a proper geometry ensuring zero microporosity and minimal chemical and banding segregations cannot be achieved. It can be seen that the segregation manifests itself at a greater scale as the time of solidification grows. A long period of solidification also encourages the coarsening of the prior austenite grains.

In all cases, the macroporosity is negligible, and the microporosity is limited to the axis of the ingot's body, on a relatively small scale. The Suzuki criterion shows the probable regions of the A-segregation and its scale. As expected, most segregations are concentrated in the head part of the ingot. Experimental verification of the prediction of A-segregation in the future will enable defining a precise range of the Suzuki criterion for a given steel grade.

From this, it follows that the design of the new mould geometry for conical steel ingots is reliable and ready to be used in laboratory-scale metallurgical applications.

**Author Contributions:** Methodology: J.O., P.L.; data resources: J.O., P.L.; numerical modelling: J.O.; writing-original draft: J.O.; writing—review and editing: J.O., P.L., T.S., and P.M. All authors have read and agreed to the published version of the manuscript.

**Funding:** Existing research and development has been realized under the financial support of the Technological Agency of the Czech Republic within the project Research and development of austenitic stainless steel for long-term storage of spent nuclear fuel (TK01020144). The paper was further supported within the project Research of advanced steels with unique properties, No. CZ.02.1.01/0.0/0.0/16_019/0000836, financed from European Regional Development Fund (ERDF).

**Conflicts of Interest:** The authors declare no conflict of interest. The funders had no role in the design of the study; in the collection, analyses, or interpretation of data; in the writing of the manuscript; or in the decision to publish the results.

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
