# Peer review of "Development of Universal Mould Geometry for the Teeming of Cylindrical Iron-Base Alloy Ingots"

_metals, doi:10.3390/met11030471_

Round 1
Reviewer 1 Report
Merry Christmas! I have read this article and do not see anything that could characterize it as a scientific publication. It is an advertisement for the technical capabilities of commercial software. The authors did not even try to interpret and generalize the obtained data and find new patterns. They limited themselves to stating the results shown in the figures.The conclusion consists of repeating those trivial observations that follow from the simulation results and could have been done without carrying out such large-scale calculations. Half of the conclusion is the imagination of the authors, how they are going to correct the defects of cast metal during further processing. And the main thing: there is not even an attempt to verify the calculation results at least on a simple example of one steel or in comparison with the results published by other authors.
I can only write a negative review. Sorry!
Author Response
Dear reviewer,
Thank you for your review. We appreciate your relevant comments and we are aware of rather technical nature of this work. The present paper has been written for a Special Issue of Metals which focuses on numerical modelling in steel metallurgy. The problem addressed by the present paper is the profound lack of adequate set of criteria usually used for simulation of this kind. Particularly, the Suzuki criterion for A-segregation is often omitted. To the best knowledge of authors’ collective, no publication has dealt with comparison of such diverse set of materials to all relevant criteria, and its application to 500kg hematite mold. This mold size is suitable for relatively large scale laboratory purposes where flexibility together with appropriate costs are needed. Therefore we believe that it is important to publish this study in this special issue on numerical modelling in steel metallurgy. Of course, more scientific approach would include various modifications of the model. However, no relevant data are available for such purpose so far.
Best regards,
Tom
Reviewer 2 Report
Material attached.

Author Response
Dear reviewer,
Thank you for your review. We took all your comments into account. As for comment no. 3, we are aware of metallurgical aspects being significant parameters of a great importance. However, thermodynamic database can be easily generated for all presented materials (based on chemical composition) in simulation software like JMatPro, ThermoCalc, etc. Moreover, amount of data for all combination of materials would be just huge for the purposes of such article. The mold is made of standard cast iron with lamellar graphite. We included this remark in the text.
Best regards,
Tom
Reviewer 3 Report
Dear Authors,
The paper sent to review will be interested for foundries plants employees who use simulation software. Moreover, the paper will be interested for the "Metals" paper readers especially for Special Issue Numerical Modelling in Steel Metallurgy.
The Authors applied the most popular and recognized as the best simulation code named MAGMA. The reviewed paper should be considered as being at a good level and follows current trends in the area of simulation. Title and keywords are appropriate and adequate to paper content. The results are presented comprehensively. The literature review is a little out-of-date. The references list should be extended by few papers from 2018-2020. The goals of the paper are sufficiently explained. The conclusions are adequate.
However, there were found some issues which require additional correction. They were listed below":
- unification of naming: mould - mold; I state that better word will be mold,
- line 3, remove full stop at the and of title,
- all acronyms in abstract should be explained even so popular like FEM,
- I suggest to remove the last sentence in abstract (line 23) or move it to the introduction; it is well known that the simulation does not always match the experiment and it is good to make validation what was not done in this work,
- line 38, is 4,25 should be 4.25,
- line 46, is author's should be authors',
- line 56, is on Figures should be in Figures,
- there are no references to literature in the introduction. The introduction is more like a methodology,
- line 96, is Cutaway better term is cross-section,
- line 126-127: one should also consider the curve of the solid fraction (FS), which affects on the contribution of porosity,
- Table 2, instead of Teeming should be Initial temperature,
- Table 2, is [deg.C] should be (deg.C) or unified to the rest,
- line 177-178, the text is in Czech and must be removed, it was not eliminated after translation (one can find the translation in next sentence),
- line 233, is Figure 5 should be Figure 7,
- line 233, is Figure 6 should be Figure 8
Best regards
Author Response
Dear reviewer,
Thank you for your review. We took all your comments into account. Regarding the solid fraction curves, they are of a great importance, of course. However, we find them counter-productive for the purposes of this article. There would be just too many (probably misleading) figures showing huge amount of data. Solid fraction curves would significantly differ in both radial and lateral directions. The literature might seem a little out-of-date. However, to the best knowledge of authors’ collective, no publication has dealt with comparison of such diverse set of materials to all relevant criteria, and its application to 500kg hematite mold. This mold size is suitable for relatively large scale laboratory purposes where flexibility together with appropriate costs are needed. Therefore we believe that it is important to publish this study in this special issue on numerical modelling in steel metallurgy. Moreover, we believe that there is no point in trying to cite “the newest possible” works. All relevant citations are stated.
Best regards,
Tom
Reviewer 4 Report
The authors develop a mould geometry suitable for casting both low and high-alloy steel grades into 500kg ingots. The work has a high application value. But some comments may be for your reference.
- Model validation should be necessary before drawing conclusions.
- Correlation between different steel grades should be clarifed, which makes the paper more systematic.
- Quantitative conclusions may be better after optimisation.
Author Response
Dear reviewer,
Thank you for your review. We took all your comments into account. Of course, more scientific approach would include validation and modifications of the model. However, no relevant data (or, very limited set) are available for such purpose. We chose these steel grades as a representative set of troublesome materials with very different behaviour from the casting point of view. We have experience with those steels and we will be able to validate our data in the future on the basis of real experiments.
Best regards,
Tom
Round 2
Reviewer 1 Report
The article still demonstrates the capabilities of commercial software Magma Soft only and does not contain a scientific discussion about the subject of a research.
In order to return to the question of the publication possibility, the authors should correct at least the following points.
- 300-305. This is the second half of the Conclusion, consisting of the authors fantasies, which are in no way based on the obtained simulation results. These arguments are admissible in the text of the article, but they should be removed from the Conclusion.
- 18-22. The abstract describes what exactly was the subject of modeling (A-segregations, PDAS, SDAS) and what “Those material features present valuable information for an estimate of material behavior in terms of crack initiation and propagation in subsequent hot forming and heat treatment processes”.
There are neither results nor discussion about "crack initiation and propagation ..." in the text of the article therefore this phrase should be removed from the abstract.
- 55-57. The authors should explain, referring to the literature, how the dendritic structure is related to the size of the primary austenite grain?
The authors investigated too different steels to answer this question in one phrase.
It all depends on the crystallization mechanism: through delta ferrite or through austenite. In the latter case, one dendrite is one austenite grain, and here the relationship between the dendritic structure and the size of the austenite grain is obvious. The situation is quite different with steels crystallizing through delta ferrite. Austenite is formed by ignoring the dendritic structure! The authors should divide the studied steels into these 2 classes and explain what will be the relationship between the dendritic and grain structure of the cast metal in each case.
- The only novelty of the article, which the authors pointed to, is the accounting for A-segregations. However, it is completely incomprehensible whether this kind of segregation is relevant for specific steels, taking into account the rather small weight of the ingot and the extremely low sulfur content in the steels under study (0.001%).
- The value of the simulation results can be assessed only after model verification. The authors don't even mention it!
- The calculation results should be summarized on the graphs, from which it clearly follows how the optimization of the ingot geometry was made on the basis of porosity, segregations, PDAS, SDAS obtained during the modeling.
- 99-154. When presenting "Numerical Model" authors should indicate where is the contribution of the authors, and where is the theory borrowed from the software user manual.
The above notes are the minimum that must be corrected for publishing an article.
Author Response
Dear Reviewer,
thank you for your comments and recommendations. We agree that the Abstract contained misleading formulations. We corrected it. Please, see our response to all your comments below.
The paper is focused on the design of a universal mould that should facilitate the production of small pre-production ingots made of a great variety of steels, and with sufficient internal quality. New equations and fine tuning of the model is far beyond the scope of this paper. It supposed to be more technical with the results that could be ready to use. Again, the main goal was to provide a reader a useful result, presenting a compromise between the porosity, the segregation, PDAS and SDAS. The greatest asset of this work is the value of H/D ratio and other mould parameters, that can be appreciated by both scientific and commercial subjects.
The relation between the dendritic structure and the PAGS is well-known and it belongs to the basic knowledge concerning this topic.
Describing the mechanism of delta-ferrite formation is irrelevant since it is an unwanted structure and for the particular steel, it is usually being avoided using the Schaeffler-Delong diagram. In any case, the primary austenite grain is of great importance and it is always taken into account. Especially, when considering high deformation resistance of the austenite. One has to admit the statement, the finer the primary structure, the easier the hot-working.
Again, repeating of basic knowledge is beyond the scope of this paper. As for A-segregation, they do manifest themselves even in smaller ingots than discussed in this paper. And the formation is a function of the mould geometry, heat transfer, and other elements, not only sulphur.
Authors of this paper clearly stated that the mould will be tested in 1Q/2021. FYI, the first test is already scheduled for February.
From our point of view, the results presented in the form of graph might be misleading.
From our point of view, we cited a relevant theory in reasonable manner and our contribution can be readily seen.
For authors’ collective,
Tomáš Studecký
Round 3
Reviewer 1 Report
Dear Authors,
Please find the comments on your answers in the attached file. Comments are written in blue.
Thanks.
